# A Deep Modality-Specific Ensemble for Improving Pneumonia Detection in Chest X-rays

**DOI:** 10.3390/diagnostics12061442

**Published:** 2022-06-11

**Authors:** Sivaramakrishnan Rajaraman, Peng Guo, Zhiyun Xue, Sameer K. Antani

**Affiliations:** Computational Health Research Branch, National Library of Medicine, National Institutes of Health, Bethesda, MD 20894, USA; peng.guo@nih.gov (P.G.); zhiyun.xue@nih.gov (Z.X.); santani@mail.nih.gov (S.K.A.)

**Keywords:** chest X-ray, deep learning, modality-specific knowledge, object detection, RetinaNet, ensemble learning, pneumonia, mean average precision

## Abstract

Pneumonia is an acute respiratory infectious disease caused by bacteria, fungi, or viruses. Fluid-filled lungs due to the disease result in painful breathing difficulties and reduced oxygen intake. Effective diagnosis is critical for appropriate and timely treatment and improving survival. Chest X-rays (CXRs) are routinely used to screen for the infection. Computer-aided detection methods using conventional deep learning (DL) models for identifying pneumonia-consistent manifestations in CXRs have demonstrated superiority over traditional machine learning approaches. However, their performance is still inadequate to aid in clinical decision-making. This study improves upon the state of the art as follows. Specifically, we train a DL classifier on large collections of CXR images to develop a CXR modality-specific model. Next, we use this model as the classifier backbone in the RetinaNet object detection network. We also initialize this backbone using random weights and ImageNet-pretrained weights. Finally, we construct an ensemble of the best-performing models resulting in improved detection of pneumonia-consistent findings. Experimental results demonstrate that an ensemble of the top-3 performing RetinaNet models outperformed individual models in terms of the mean average precision (mAP) metric (0.3272, 95% CI: (0.3006,0.3538)) toward this task, which is markedly higher than the state of the art (mAP: 0.2547). This performance improvement is attributed to the key modifications in initializing the weights of classifier backbones and constructing model ensembles to reduce prediction variance compared to individual constituent models.

## 1. Introduction

Pneumonia is an acute respiratory infectious disease that can be caused by various pathogens such as bacteria, fungi, or viruses [1]. The infection affects the alveoli in the lungs by filling them up with fluid or pus, thereby resulting in reduced intake of oxygen and causing difficulties in breathing. The potency of the disease depends on several factors including age, health, and the source of infection. According to the World Health Organization (WHO) report (https://www.who.int/news-room/fact-sheets/detail/pneumonia, accessed on 11 December 2021), pneumonia is reported to be an infectious disease that results in a higher mortality rate, particularly in children. About 22% of all deaths in pediatrics from 1 to 5 years of age are reported to result from this infection. Effective diagnosis and treatment of pneumonia are therefore critical to improving patient care and survival rate.

Chest X-rays (CXRs) are commonly used to screen for pneumonia infection [2,3]. Analysis of CXR images can be particularly challenging in low and middle-income countries due to a lack of expert resources, socio-economic factors, etc. [4]. Computer-aided detection systems using conventional deep learning (DL) methods, a sub-class of machine learning (ML) algorithms can alleviate this burden and have demonstrated superiority over traditional machine learning methods in detecting disease regions of interest (ROIs) [5,6]. Such algorithms (i) automatically detect pneumonia-consistent manifestations on CXRs; and (ii) can support clinical-decision making by facilitating swift referrals for critical cases to improve patient care.

### 1.1. Related Works

A study of the literature reveals several studies that propose automated methods using DL models for detecting pneumonia-consistent manifestations on CXRs. However, DL models vary in their architecture and learn discriminative features from different regions in the feature space. They are observed to be highly sensitive to data fluctuations resulting in poor generalizability due to varying degrees of biases and variances. An approach to achieving a low bias and variance and ensuring reliable outcomes is using ensemble learning which is an established ML paradigm that combines predictions from multiple diverse DL models and improves performance compared to individual constituent models [7]. The authors of [8] proposed an ensemble of Faster-RCNN [9], Yolov5 [8], and EfficientDet [8] models to localize and predict bounding boxes containing pneumonia-consistent findings in the publicly available VinDr-CXR [8] dataset and reported a mean Average Precision (mAP) of 0.292. The following methods used ensembled object detection models to detect pneumonia-consistent findings using the CXR collection hosted for the RSNA Kaggle pneumonia detection challenge (https://www.kaggle.com/c/rsna-pneumonia-detection-challenge accessed on 3 March 2022). The current state-of-the-art method according to the challenge leaderboard (https://www.kaggle.com/competitions/rsna-pneumonia-detection-challenge/leaderboard accessed on 3 March 2022) has a mAP of 0.2547. In [10], an ensemble of RetinaNet [11] and Mask RCNN models with ResNet-50 and ResNet-101 classifier backbones delivered a performance with a mAP of 0.2283 using the RSNA Kaggle pneumonia detection challenge CXR dataset. Another study [12] proposed a weighted-voting ensemble of the predictions from Mask R-CNN and RetinaNet models to achieve an mAP of 0.2174 in detecting pneumonia-consistent manifestations. These studies used the randomized test set split from the challenge-provided training data. This is a serious concern since the organizers have not made the blinded test set used during the challenge available for further use. This cripples follow-on research, such as ours, from making fair comparisons.

### 1.2. Rationale for the Study

All above studies used off-the-shelf DL object detection models with ImageNet [13] pretrained classifier backbones. However, ImageNet is a collection of stock photographic images whose visual characteristics, including shape and texture among others, are distinct from CXRs. As well, the disease-specific ROIs in CXRs are relatively small and many go unnoticed which may result in suboptimal predictions [14]. Our prior works and other literature have demonstrated that the knowledge transferred from DL models that are retrained on a large collection of CXR images is shown to improve performance on relevant target medical visual recognition tasks [15,16,17]. To the best of our knowledge, we observed that no literature discussed the use of CXR modality-specific backbones in object detection models, particularly applied to detecting pneumonia-consistent findings in CXRs.

### 1.3. Contributions of the Study

Our study improves upon the state-of-the-art as follows:(i).To the best of our knowledge, this is the first study that studies the impact of using CXR modality-specific classifier backbones in a RetinaNet-based object detection model, particularly applied to detecting pneumonia-consistent findings in CXRs.(ii).We train state-of-the-art DL classifiers on large collections of CXR images to develop CXR modality-specific models. Next, we use these models as the classifier backbone in the RetinaNet object detection network. We also initialize this backbone using random weights and ImageNet-pretrained weights to compare detection performance. Finally, we construct an ensemble of the aforementioned models resulting in improved detection of pneumonia-consistent findings.(iii).Through this approach, we aim to study the combined benefits of various weight initializations for classifier backbones and construct an ensemble of the best-performing models to improve detection performance. The models’ performance is evaluated in terms of mAP and statistical significance is reported in terms of confidence intervals (CIs) and *p*-values.

Section 2 discusses the datasets, model architecture, training strategies, loss functions, evaluation metrics, statistical methods, and computational resources, Section 3 elaborates on the results and Section 4 concludes this study.

## 2. Materials and Methods

### 2.1. Data Collection and Preprocessing

The following data collections are used for this study:(i).CheXpert CXR [18]: The dataset includes 223,648 frontal and lateral CXR images that are collected from 65,240 patients at Stanford Hospital, California, USA. The CXRs are labeled for 14 cardiopulmonary disease manifestations, the details are extracted from the associated radiology reports using an automated labeling algorithm.(ii).TBX11K CXR [19]: This collection includes 11,200 CXRs collected from normal patients and those with other cardiopulmonary abnormalities. The abnormal CXRs are collected from patients tested with the microbiological gold standard. There are 5000 CXRs showing no abnormalities and 6200 CXRs showing other abnormal findings including those collected from sick patients (n = 5000), active Tuberculosis (TB) (n = 924), latent Tuberculosis (n = 212), active and latent TB (n = 54), and other uncertain (n = 10) cases. The regions showing TB-consistent manifestations are labeled for the abnormal regions using coarse rectangular bounding boxes.(iii).RSNA CXR [20]: This CXR collection is released by RSNA for the RSNA Kaggle Pneumonia detection challenge. The collection consists of 26,684 CXRs that include 6012 CXR images showing pneumonia-consistent manifestations, 8851 CXRs showing no abnormal findings, and 11,821 CXRs showing other cardiopulmonary abnormalities. The CXRs showing pneumonia-consistent findings are labeled for abnormal regions using rectangular bounding boxes and are made available for the detection challenge.

We used the frontal CXRs from the CheXpert and TBX11K data collection during CXR image modality-specific retraining and those from the RSNA CXR collection to train the RetinaNet-based object detection models. All images are resized to 512 × 512 spatial dimensions to reduce computation complexity. The contrast of the CXRs is further increased by saturating the top 1% and bottom 1% of all the image pixel values. For CXR modality-specific retraining, the frontal CXR projections from the CheXpert and TBX11K datasets are divided at the patient level into 70% for training, 10% for validation, and 20% for testing. This patient-level split prevents the leakage of data and subsequent bias during model training. For object detection, the frontal CXRs from the RSNA CXR dataset that shows pneumonia-consistent manifestations are divided at the patient level into 70% for training, 10% for validation, and 20% for testing. Table 1 shows the number of CXR images across the training, validation, and test sets used for CXR modality-specific retraining and object detection, respectively.

### 2.2. Model Architecture

#### 2.2.1. CXR Modality-Specific Retraining

The ImageNet-pretrained DL models, viz., VGG-16, VGG-19, DenseNet-121, ResNet-50, EfficientNet-B0, and MobileNet have demonstrated promising performance in several medical visual recognition tasks [14,19,21,22,23]. These models are further retrained on a large collection of CXR images to classify them as showing cardiopulmonary abnormal manifestations or no abnormalities. Such retraining helps the models to learn CXR image modality-specific features that can be transferred and fine-tuned to improve performance in a relevant task using CXR images. The best-performing model with the learned CXR image modality-specific weights is used as the classifier backbone to train the RetinaNet-based object detection model toward detecting pneumonia-consistent manifestations. Figure 1 shows the block diagram illustrating the steps involved in CXR image modality-specific retraining.

#### 2.2.2. RetinaNet Architecture

We used RetinaNet as the base object detection architecture in our experiments. The architecture of the RetinaNet model is shown in Figure 2. As a single-stage object detection structure, RetinaNet shares a similar concept of “anchor proposal” with [24]. It used a feature pyramid network (FPN) [25] where features on each of the image scales are computed separately in the lateral connections and then summed up through convolutional operations via the top-down pathways. The FPN network combines low-resolution features with strong semantic information, and high-resolution features with weak semantics through top-down paths and horizontal connections. Thus, feature maps with rich semantic information are obtained that would prove beneficial for detecting relatively smaller ROIs consistent with pneumonia compared to the other parts of the CXR image. Furthermore, when trained to minimize the focal loss [5], the RetinaNet was reported to deliver significant performance focusing on hard, misclassified examples.

#### 2.2.3. Ensemble of RetinaNet Models with Various Backbones

We initialized the weights of the VGG-16 and ResNet-50 classifier backbones used in the RetinaNet model using three strategies: (i) Random weights; (ii) ImageNet-pretrained weights, and (iii) CXR image modality-specific retrained weights as discussed in Section 2.2.1. Each model is trained for 80 epochs and the model weights (snapshots) are stored at the end of each epoch. Varying modifications of the RetinaNet model classifier backbones and loss functions are mentioned in Table 2.

We adopted the non-maximum suppression (NMS) in the RetinaNet training with an IoU threshold of 0.5 and evaluated the models using all the predictions with a confidence score over 0.9. A weighted averaging ensemble is constructed using (i) the top-3 performing models from the 12 RetinaNet models mentioned in Table 2, and (ii) the top-3 performing snapshots (model weights) using each classifier backbone. We empirically assigned the weights as 1, 0.9, and 0.8 for the predictions of the 1st, 2nd, and 3rd best performing models. A schematic of the ensemble procedure is shown in Figure 3. An ensembled bounding box is generated if the IOU of the weighted average of the predicted bounding boxes and the ground truth (GT) boxes is greater than 0.5. The ensembled model is evaluated based on the mean average precision (mAP) metric.

#### 2.2.4. Loss Functions and Evaluation Metrics

##### CXR Image Modality-Specific Retraining

During CXR image modality-specific retraining, the DL models are retrained on a combined selection of the frontal CXR projections from the CheXpert and TBX11K datasets (details in Table 1). The training is performed for 128 epochs to minimize the categorical cross-entropy (CCE) loss. The CCE loss is the most commonly used loss function in classification tasks, and it helps to measure the distinguishability between two discrete probability distributions. It is expressed as shown in Equation (1).
(1)CCEloss=−∑k=1output sizeyklogyk^

Here, yk^ denotes the *k*th scalar value in the model output, $y_{k}$ denotes the corresponding target, and the output size denotes the number of scalar values in the model output. The term $y_{k}$ denotes the probability that event *k* occurs and the sum of all $y_{k}=1$. The minus sign in the CCE loss equation ensures the loss is minimized when the distributions become less distinguishable. We used a stochastic gradient descent optimizer with an initial learning rate of 1 × 10^−4^ and momentum of 0.9 to reduce the CCE loss and improve performance. Callbacks are used to store the model checkpoints and the learning rate is reduced after a patience parameter of 10 epochs when the validation performance ceased to improve. The weights of the model that delivered a superior performance with the validation set are used to predict the test set. The models are evaluated in terms of accuracy, the area under the receiver-operating characteristic curve (AUROC), the area under the precision-recall (PR) curve (AUPRC), sensitivity, precision, F-score, Matthews correlation coefficient (MCC), and Kappa statistic. 

##### RetinaNet-Based Detection of Pneumonia-Consistent Findings

Considering medical images, the disease ROIs span a relatively smaller portion of the whole image. This results in a considerably high degree of imbalance in the foreground ROI and the background pixels. These issues are particularly prominent in applications such as detecting cardiopulmonary manifestations like pneumonia where the number of pixels showing pneumonia-consistent manifestations is markedly lower compared to the total number of image pixels. Generalized loss functions such as balanced cross-entropy loss do not take this data imbalance into account. This may lead to a learning bias and subsequent adversity in learning the minority ROI pixels. Appropriate selection of the loss function is therefore critical for improving detection performance. In this regard, the authors of [11] proposed the focal loss for object detection, an extension of the cross-entropy loss, which alleviates this learning bias by giving importance to the minority ROI pixels while down-weighting the majority background pixels. Minimizing the focal loss thereby reduces the loss contribution from majority background examples and increases the importance of correctly detecting the minority disease-positive ROI pixels. The focal loss is expressed as shown in Equation (2).
(2)$$Focal\;loss\left(p_{t}\right)=-\alpha_{t}{\left(1-p_{t}\right)}^{\gamma }\log\left(p_{t}\right)$$

Here, $p_{t}$ denotes the probability the object detection model predicts for the GT. The parameter *γ* decides the rate of down-weighting the majority (background non-ROI) samples. The equation converges to the conventional cross-entropy loss when *γ* = 0. We empirically selected the value of *γ* = 2 which delivered superior detection performance.

Another loss function called the Focal Tversky loss function [27], a generalization of the focal loss function, is proposed to tackle the data imbalance problem and is given in Equation (3). The Focal Tversky loss function generalizes the Tversky loss which is based on the Tversky index that helps achieve a superior tradeoff between recall and precision when trained on class-imbalanced datasets. The Focal Tversky loss function uses a smoothing parameter *γ* that controls the non-linearity of the loss at different values of the Tversky index to balance between the minority pneumonia-consistent ROI and majority background classes. In Equation (3), TI denotes the Tversky index, expressed as shown in Equation (4).
(3)$$FT_{{loss}_{c}}=\underset{c}{\sum }1-TI_{c}^{\gamma }$$
(4)TIc=∑i=1Mticgic+∈∑i=1Mticgic+α ∑i=1Mtic^gic+β ∑i=1Mticgic^+∈

Here, $g_{ic}$ and $t_{ic}$ denote the ground truth and predicted labels for the pneumonia class *c*, where $g_{ic}$ and $t_{ic}$ ∈ {0,1}. That is, $t_{ic}$ denotes the probability that the pixel *i* belongs to the pneumonia class *c* and tic^ denotes the probability that the pixel *i* belongs to the background class c^. The same holds for $g_{ic}$ and gic^. The term *M* denotes the total number of image pixels. The term ∈ provides numerical stability to avoid divide-by-zero errors. The hyperparameters *α* and *β* are tuned to emphasize recall under class-imbalanced training conditions. The Tversky index is adapted to a loss function by minimizing $\underset{c}{\sum }1-TI_{c}$. After empirical evaluations, we fixed the value of γ = 4/3, α = 0.7 and β=0.75.

As is known, the loss function within RetinaNet is a summation of a couple of loss functions, one for classification and the other for bounding box regression. We left the Smooth-L1 loss that is used for bounding box regression unchanged. For classification, we explored the performance with focal loss and focal Tversky loss functions individually for training the RetinaNet models with varying weight initializations. We used the bounding box annotations [20] associated with the RSNA CXRs showing pneumonia-consistent manifestations as the GT bounding boxes and measured its agreement with that generated by the models initialized with random weights, ImageNet-pretrained, and CXR image modality-specific retrained classifier backbones. Let TP, FP, and FN denote the true positives, false positives, and false negatives, respectively. Given a pre-defined IOU threshold, a predicted bounding box is considered to be TP if it overlaps with the GT bounding box by a value equal to or exceeding this threshold. FP denotes that the predicted bounding box has no associated GT bounding box. FN denotes the GT bounding box has no associated predicted bounding box. The mAP is measured as the area under the precision-recall curve (AUPRC) as shown in Equation (5). Here, *P* denotes precision which measures the accuracy of predictions, and *R* denotes recall which measures how well the model identifies all the TPs. They are computed as shown in Equations (6) and (7). The value of mAP lies in the range [0, 1].
(5)$$mean\;average\;precision\;\left(mAP\right)=\int_{0}^{1}P\left(R\right)dR$$
(6)$$Precision\;\left(P\right)=\frac{TP}{TP+FP}$$
(7)$$Recall=\frac{TP}{\left(TP+FN\right)}$$

We used a Linux system with 1080Ti GPU, the Tensorflow backend (v. 2.6.2) with Keras, and CUDA/CUDNN libraries for accelerating the graphical processing unit (GPU) toward training the object detection models that are configured in the Python environment.

### 2.3. Statistical Analysis

We evaluated statistical significance using the mAP metric achieved by the models trained with various weight initializations and loss functions. The 95% confidence intervals (CIs) are measured as the binomial interval using the Clopper-Pearson method.

## 3. Results and Discussion

We organized the results from our experiments into the following sections: Evaluating the performance of (i) CXR image modality-specific retrained models and (ii) RetinaNet object detection models using classifier backbones with varying weight initializations and loss functions.

### 3.1. Classification Performance during CXR Image Modality-Specific Retraining

Recall that the ImageNet-pretrained DL models are retrained on the combined selection of CXRs from the CheXpert and TBX11K collection. Such retraining is performed to convert the weight layers specific to the CXR image modality and let the models learn CXR modality-specific features to improve performance when the learned knowledge is transferred and fine-tuned for a related medical image visual recognition task. The performance achieved by the CXR image modality-specific retrained models using the hold-out test set is listed in Table 3 and the performance curves are shown in Figure 4. The *no-skill* line in Figure 4 denotes the performance when a classifier would fail to discriminate between the normal and abnormal CXRs and therefore would predict a random outcome or a specific category under all circumstances.

We could observe from Table 3 that the CXR image modality-specific retrained VGG-16 model demonstrates the best performance compared to other models in terms of all metrics except sensitivity. Of these, the MCC metric is a good measure to use because unlike F-score because it considers a balanced ratio of TPs TNs, FPs, and FNs. We noticed that the differences in the MCC values achieved by the various CXR image modality-specific retrained models are not significantly different (*p* > 0.05). Based on its performance, we used VGG-16 as the backbone for the RetinaNet detector. However, to enable fair comparison with other conventional RetinaNet-based results, we included the ResNet-50 backbone for detecting pneumonia-consistent manifestations. The VGG-16 and ResNet-50 classifier backbones are also initialized with random and ImageNet-pretrained weights for further comparison.

### 3.2. Detection Performance Using RetinaNet Models and Their Ensembles

Recall that the RetinaNet models are trained with different initializations of the classifier backbones. The performance achieved by these models using the hold-out test set is listed in Table 4. Figure 5 shows the PR curves obtained with the RetinaNet model using varying weight initializations for the selected classifier backbones. These curves show the precision and recall value of the model’s bounding box predictions on every sample in the test set. We observe from Table 4 that the RetinaNet model with the CXR image modality-specific retrained ResNet-50 classifier backbone and trained using the focal loss function demonstrates superior performance in terms of mAP. Figure 6 shows the bounding box predictions of the top-3 performing RetinaNet models for a sample CXR from the hold-out test set.

We used two approaches to combine the bounding box predictions. They are (i) using the bounding box predictions from the top-3 performing RetinaNet models, viz., ResNet-50 with CXR image modality-specific weights + focal loss, ResNet-50 with CXR image modality-specific weights + focal Tversky loss, and ResNet-50 with random weights + focal loss; and, (ii) using the bounding box predictions from the top-3 performing snapshots (weights) within each model. The results are presented in Table 5 and Figure 7. A weighted averaging ensemble of the bounding boxes is generated when the IoU of the predicted bounding boxes is greater than the threshold value which is set at 0.5. Recall that the models are trained for 80 epochs and a snapshot (i.e., the model weights) is stored at the end of each epoch. We observed that the ensemble of the top-3 performing RetinaNet models delivered superior performance in terms of mAP metric compared to other models and ensembles. Figure 8 shows a sample CXR image with GT and predicted bounding boxes using the weighted averaging ensemble of the top-3 individual models and the top-3 snapshots of the best-performing model.

## 4. Conclusions and Future Work

In this study, we demonstrated the combined benefits of training CXR image modality-specific models, using them as backbones in an object detection model, evaluating them in different loss settings, and constructing ensembles of the best-performing models to improve performance in a pneumonia detection task. We observed that both CXR image modality-specific classifier backbones and ensemble learning improved detection performance compared to the individual constituent models. This study, however, suffers from the limitation that we have only investigated the effect of using CXR modality-specific classifier backbones in a RetinaNet-based object detection model to improve detecting pneumonia-consistent findings. The efficacy of this approach in detecting other cardiopulmonary disease manifestations is a potential avenue for future research. Additional diversity in the training process could be introduced by using CXR images and their disease-specific annotations collected from multiple institutions. With the advent of high-performance computing and current advancements in DL-based object detection, future studies could explore the use of mask x-RCNN, transformer-based models, and other advanced detection methods [28,29,30,31] and their ensembles in improving detection performance. Novel model optimization methods and loss functions can be proposed to further improve detection performance. However, the objective of this study is not to propose a new objection detection model but to validate the use of CXR modality-specific classifier backbones in existing models to improve performance. As the organizers of the RSNA Kaggle pneumonia detection challenge have not made the blinded GT annotations of the test set publicly available, we are unable to compare our results with the challenge leaderboard. However, the performance of our method on a random split from the challenge-provided training set, where we sequester 10% of the images for testing, using 70% for training and 20% for validation, respectively, is markedly superior to the best performing method on the leaderboard.

## Figures and Tables

**Figure 1 diagnostics-12-01442-f001:**
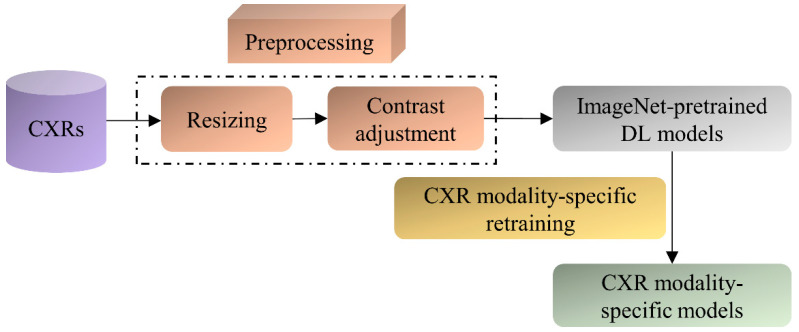
Steps illustrating CXR image modality-specific retraining of the ImageNet-pretrained models.

**Figure 2 diagnostics-12-01442-f002:**
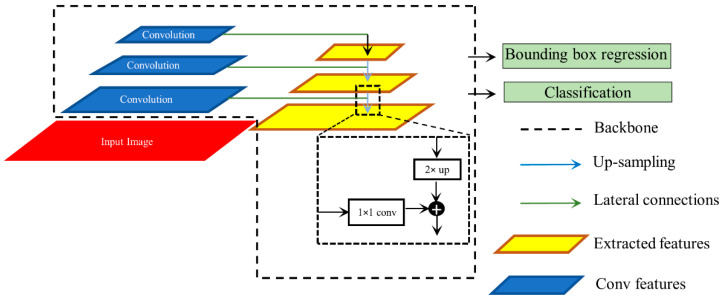
Method flowchart for the RetinaNet network.

**Figure 3 diagnostics-12-01442-f003:**
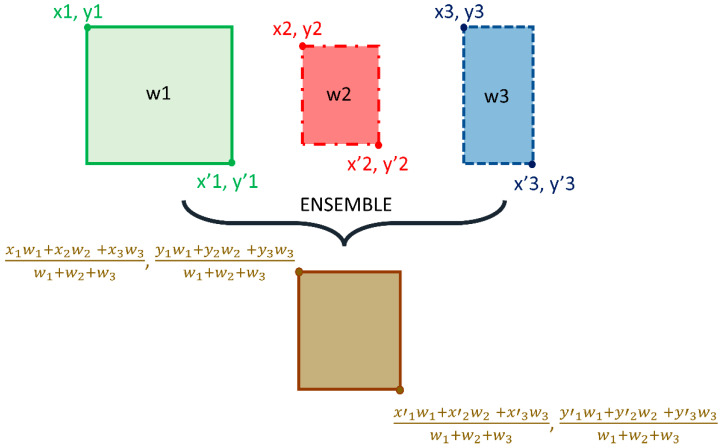
Method Schematic of the ensemble approach.

**Figure 4 diagnostics-12-01442-f004:**
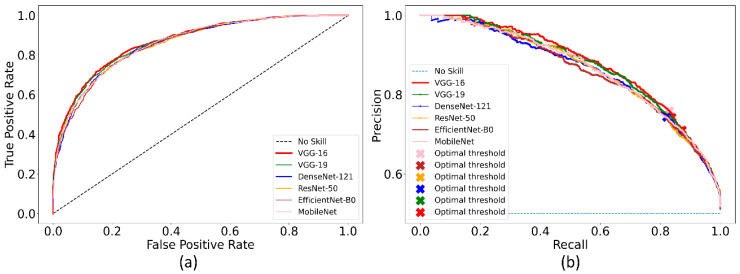
The collection of performance curves for the CXR image modality-specific retrained models. The performance is recorded at the optimal classification threshold measured with the validation data. (**a**) ROC and (**b**) PR curves.

**Figure 5 diagnostics-12-01442-f005:**
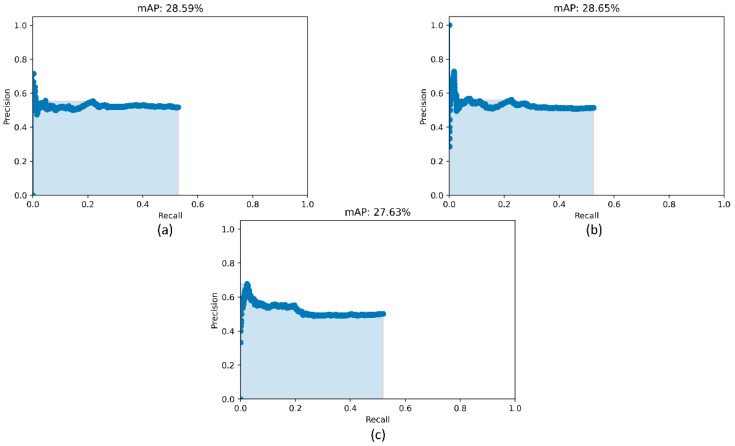
PR curves of the RetinaNet models initialized with varying weights for the classifier backbones. (**a**) ResNet-50 with CXR image modality-specific weights + focal Tversky loss; (**b**) ResNet-50 with CXR image modality-specific weights + focal loss, and (**c**) ResNet-50 with random weights + focal loss.

**Figure 6 diagnostics-12-01442-f006:**
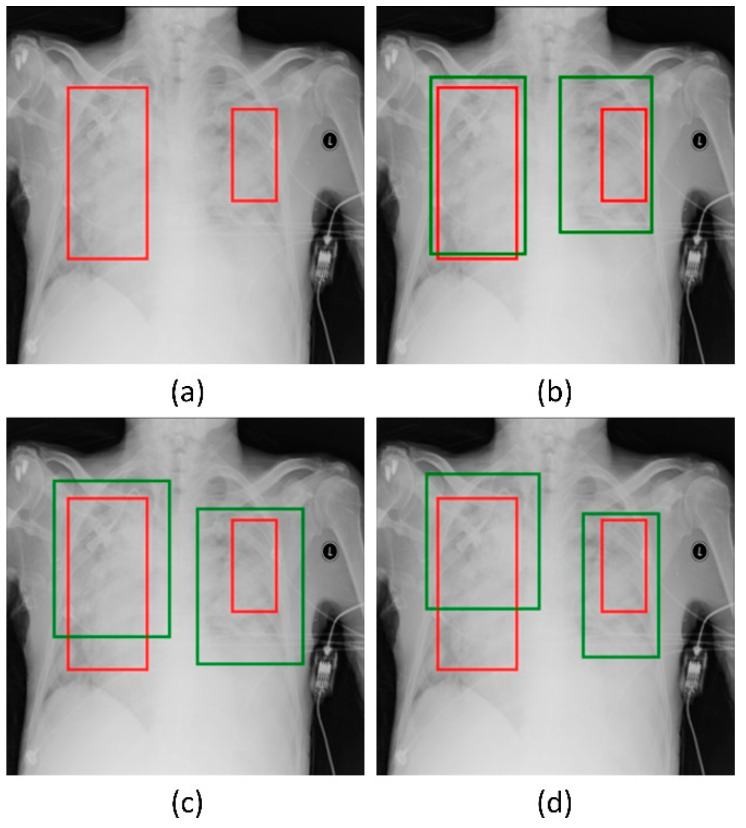
Bounding box predictions of the RetinaNet models initialized with varying weights for the classifier backbones. Green boxes denote the model predictions and red boxes denote the ground truth. (**a**) A sample CXR with ground truth bounding boxes. (**b**) ResNet-50 with CXR image modality-specific weights + focal Tversky loss; (**c**) ResNet-50 with CXR image modality-specific weights + focal loss, and (**d**) ResNet-50 with random weights + focal loss.

**Figure 7 diagnostics-12-01442-f007:**
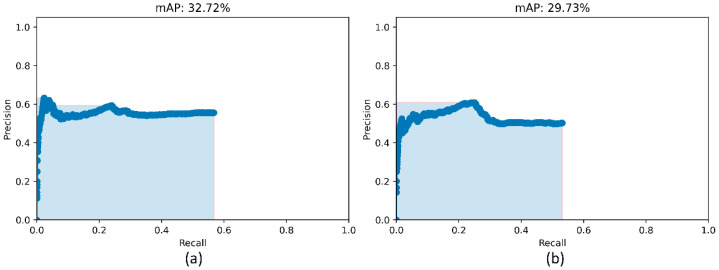
PR curves of the model ensembles. (**a**) PR curve obtained with the weighted-averaging ensemble of top-3 performing models (ResNet-50 with CXR modality-specific weights + focal loss, ResNet-50 with CXR modality-specific weights + focal Tversky loss, and ResNet-50 with random weights + focal loss and (**b**) PR curve obtained with the ensemble of top-3 performing snapshots while training the ResNet-50 with CXR modality-specific weights + focal loss model.

**Figure 8 diagnostics-12-01442-f008:**
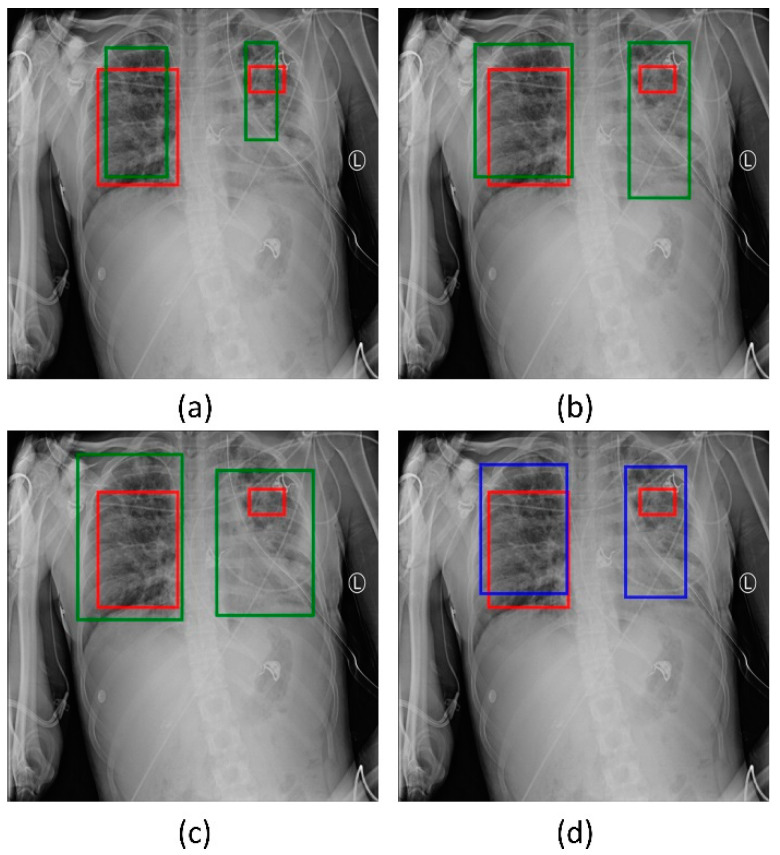
Bounding box predictions using the ensemble of RetinaNet models initialized with varying weights for the classifier backbones. Green boxes denote the individual model predictions, blue boxes denote the ensemble predictions and red boxes denote the ground truth. (**a**) ResNet-50 with CXR image modality-specific weights + focal Tversky loss; (**b**) ResNet-50 with CXR image modality-specific weights + focal loss; (**c**) ResNet-50 with random weights + focal loss, and (**d**) the ensembled bounding box prediction.

**Table 1 diagnostics-12-01442-t001:** Patient-level dataset splits show the number of images for CXR modality-specific retraining and object detection. Note: TBX11K and RSNA datasets have one image per patient.

Dataset	Train		Validation		Test	
	Abnormal	Normal	Abnormal	Normal	Abnormal	Normal
CXR Modality-specific retraining						
CheXpert	13,600	13,600	1700	1700	1700	1700
TBX11k	3040	3040	380	380	380	380
RetinaNet-based object detection						
Dataset	Train		Validation		Test	
RSNA	4212		600		1200	

**Table 2 diagnostics-12-01442-t002:** RetinaNet model classifier backbones with varying weight initializations and loss functions. The loss functions mentioned are used for classification. For bounding box regression, only the smooth-L1 loss function [26] is used in all cases.

ResNet-50 Backbone and Classification Loss Functions	VGG-16 Backbone and Classification Loss Functions
ResNet-50 with random weights + focal loss	VGG-16 with random weights + focal loss
ResNet-50 with random weights + focal Tversky loss	VGG-16 with random weights + focal Tversky loss
ResNet-50 with ImageNet pretrained weights + focal loss	VGG-16 with ImageNet pretrained weights + focal loss
ResNet-50 with ImageNet pretrained weights + focal Tversky loss	VGG-16 with ImageNet pretrained weights + focal Tversky loss
ResNet-50 with CXR image modality-specific weights + focal loss	VGG-16 with CXR image modality-specific weights + focal loss
ResNet-50 with CXR image modality-specific weights + focal Tversky loss	VGG-16 with CXR image modality-specific weights + focal Tversky loss

**Table 3 diagnostics-12-01442-t003:** Performance of the CXR image modality-specific retrained models with the hold-out test set. Bold numerical values denote superior performance. The values in parenthesis denote the 95% CI for the MCC metric.

Models	Accuracy	AUROC	AUPRC	Sensitivity	Precision	F-Score	MCC	Kappa
VGG-16	**0.7834**	**0.8701**	**0.8777**	0.8303	**0.7591**	**0.7931**	**0.5693** **(0.5542, 0.5844)**	**0.5668**
VGG-19	0.7743	0.8660	0.8727	0.8389	0.7429	0.7880	0.5532 (0.5380, 0.5684)	0.5486
DenseNet-121	0.7738	0.8582	0.8618	0.8264	0.7477	0.7851	0.5507 (0.5355, 0.5659)	0.5476
ResNet-50	0.7685	0.8586	0.8646	0.8207	0.7431	0.7800	0.5400 (0.5248, 0.5552)	0.5370
EfficientNet-B0	0.7553	0.8568	0.8612	0.8678	0.7084	0.7800	0.5240 (0.5088, 0.5392)	0.5106
MobileNet	0.7584	0.8609	0.8655	**0.8726**	0.7104	0.7832	0.5309 (0.5157, 0.5461)	0.5168

**Table 4 diagnostics-12-01442-t004:** Performance of RetinaNet with the varying weight initializations for the classifier backbones and training losses. The values in parenthesis denote the 95% CI for the mAP metric. Bold numerical values denote superior performance.

Models	AUPRC (mAP)
ResNet-50 with random weights + focal loss	0.2763 (0.2509, 0.3017)
ResNet-50 with random weights + focal Tversky loss	0.2627 (0.2377, 0.2877)
ResNet-50 with ImageNet pretrained weights + focal loss	0.2719 (0.2467, 0.2971)
ResNet-50 with ImageNet pretrained weights + focal Tversky loss	0.2737 (0.2484, 0.2990)
ResNet-50 with CXR image modality-specific weights + focal loss	**0.2865 (0.2609, 0.3121)**
ResNet-50 with CXR image modality-specific weights + focal Tversky loss	0.2859 (0.2603, 0.3115)
VGG-16 with random weights + focal loss	0.2549 (0.2302, 0.2796)
VGG-16 with random weights + focal Tversky loss	0.2496 (0.2251, 0.2741)
VGG-16 with ImageNet pretrained weights + focal loss	0.2734 (0.2481, 0.2987)
VGG-16 with ImageNet pretrained weights + focal Tversky loss	0.2666 (0.2415, 0.2917)
VGG-16 with CXR image modality-specific weights + focal loss	0.2686 (0.2435, 0.2937)
VGG-16 with CXR image modality-specific weights + focal Tversky loss	0.2648 (0.2398, 0.2898)

**Table 5 diagnostics-12-01442-t005:** Ensemble performance with the top-3 performing models (from Table 4) and the top-3 snapshots for each of the models trained with various classifier backbones and weight initializations. Values in parenthesis denote the 95% CI for the mAP metric. Bold numerical values denote superior performance.

Ensemble Method	mAP
Top-3 model ensemble (ResNet-50 with CXR image modality-specific weights + focal loss, ResNet-50 with CXR image modality-specific weights + focal Tversky loss, and ResNet-50 with random weights + focal loss	**0.3272 (0.3006, 0.3538)**
Ensemble of the top-3 snapshots for each model	
ResNet-50 with random weights + focal loss	0.2777 (0.2523, 0.3031)
ResNet-50 with random weights + focal Tversky loss	0.2630 (0.2380, 0.2880)
ResNet-50 with ImageNet pretrained weights + focal loss	0.2788 (0.2534, 0.3042)
ResNet-50 with ImageNet pretrained weights + focal Tversky loss	0.2812 (0.2557, 0.3067)
ResNet-50 with CXR image modality-specific weights + focal loss	0.2973 (0.2714, 0.3232)
ResNet-50 with CXR image modality-specific weights + focal Tversky loss	0.2901 (0.2644, 0.3158)
VGG-16 with random weights + focal loss	0.2633 (0.2383, 0.2883)
VGG-16 with random weights + focal Tversky loss	0.2556 (0.2309, 0.2803)
VGG-16 with ImageNet pretrained weights + focal loss	0.2823 (0.2568, 0.3078)
VGG-16 with ImageNet pretrained weights + focal Tversky loss	0.2715 (0.2463, 0.2967)
VGG-16 with CXR image modality-specific weights + focal loss	0.2813 (0.2558, 0.3068)
VGG-16 with CXR image modality-specific weights + focal Tversky loss	0.2698 (0.2446, 0.2950)

## Data Availability

The data required to reproduce this study is publicly available and cited in the manuscript.

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
