# Peer review of "A Deep Modality-Specific Ensemble for Improving Pneumonia Detection in Chest X-rays"

_diagnostics, 2022, doi:10.3390/diagnostics12061442_

Round 1
Reviewer 1 Report
In this paper, the authors propose the use of modality-specific backbones in object detection models. This is interesting and significant for medical image based diagnosis.
Introduction:
· In line 59 mAP is mentioned and the abbreviation, mean Average Precision (mAP), is given in line 63.
· From line 68-71, the authors mentioned: “These studies used the randomized test set split from the challenge-provided training data. This is a serious concern since the organizers have not made the blinded test set used during the challenge available for further use. This cripples follow-on research, such as ours, from making fair comparisons.” Can the authors have the same argument for cross validation results?
Data Collection and Preprocessing:
· Does the study include all frontal and lateral CXR images? If only frontal images are used, please mention that clearly.
· The contrast of the CXRs is further increased by saturating the top 1% and bottom 1% of all the image pixel values. Can the authors please elaborate how crucial is this step? Will the results be significantly different otherwise?
RetinaNet Architecture:
· Can the authors please mention, why specifically RetinaNet as the base object detection? Is the line 158-160, the sole reason?
· In Figure 2, please mention the full form of conv. If it is convolution, then write as it is. Conv seems more casual.
Line 163: 2.2.3. Ensemble of RetinaNet Models with Various Backbones
Line 183: 2.2.3. Loss functions and evaluation metrics
Both these sections have the same number. Maybe there is a mistake. Please check.
In 2.2.3. Ensemble of RetinaNet Models with Various Backbones, it is mentioned that “Each model is trained for 80 epochs”, whereas in section 2.2.3. Loss functions and evaluation metrics it is mentioned “The training is performed for 128 epochs to minimize the cate- 187 gorical cross-entropy (CCE) loss.” Please address this more clearly.
CXR image modality-specific retraining:
· This section talks about the categorical cross-entropy (CCE) loss. But in equation (1), it is mentioned BCE. Please check if there is a mistake.
· Equation 1: Please mention in the text what is p, p’, Y, and Y’.
RetinaNet-Based Detection of Pneumonia-Consistent Findings:
· I appreciate that the authors included the following section: 2.2.3.2. RetinaNet-Based Detection of Pneumonia-Consistent Findings. This is very specific to the study and expands on the importance of several parameters for image based classification.
· In equation (3) and (5) please mention what is c, and how it is related to the algorithm architecture.
Detection Performance Using RetinaNet Models and Their Ensembles:
· Please include a schematic architecture of the ensemble concept. It would help researchers form other fields to visualize the concept more easily
· In Figure 4 and Figure 6, please mention what is AP? Is it mAP?
· Please be consistent with the image quality and size of Figure 4 and Figure 6.
General comments:
· The authors mention several times the Modality specific based model, and even in the title. CXR image modality is first mentioned in section 3.1. Please mention it earlier, if possible, in the abstract.
· Are there any overfitting, or underfitting issues related to the results?
· How accurate will it work in the presence of other respiratory Illnesses caused from infections? Can it classify between the different infections?
Author Response
In this paper, the authors propose the use of modality-specific backbones in object detection models. This is interesting and significant for medical image based diagnosis.
Author response: We render our sincere thanks to the reviewer for the insightful comments and for encouraging the resubmission of our manuscript. To the best of our knowledge and belief, we have addressed the concerns of the reviewer in the revised manuscript.
Q1: In line 59 mAP is mentioned and the abbreviation, mean Average Precision (mAP), is given in line 63
Author response: Agreed and modified in lines 61 – 62 of the revised manuscript.
Q2: From line 68-71, the authors mentioned: “These studies used the randomized test set split from the challenge-provided training data. This is a serious concern since the organizers have not made the blinded test set used during the challenge available for further use. This cripples follow-on research, such as ours, from making fair comparisons.” Can the authors have the same argument for cross validation results?
Author response: Yes. Cross-validation is performed to tune model hyperparameters. Once we use cross-validation for model tuning, it may no longer reliably be used to assess how well the model will perform on hold-out test data. So, a test set should still be held out for final evaluation.
Q3: Does the study include all frontal and lateral CXR images? If only frontal images are used, please mention that clearly.
Author response: Agreed and modified in lines 128 – 129 of the revised manuscript as shown below:
We use the frontal CXRs from the CheXpert and TBX11K data collection during CXR image modality-specific retraining and those from the RSNA CXR collection to train the RetinaNet-based object detection models.
Q4: The contrast of the CXRs is further increased by saturating the top 1% and bottom 1% of all the image pixel values. Can the authors please elaborate how crucial is this step? Will the results be significantly different otherwise?
Author response: Contrast adjustments help to improve perceptibility by enhancing the difference between the tissues, bones, and blood vessels. Such adjustments are important as they help radiologists distinguish between normal and abnormal conditions. We used an OpenCV-based custom function to enhance image contrast, similar to the “Imadjust” function in Matlab. We have observed from our previous studies that such a contrast enhancement improves performance.
Q5: Can the authors please mention, why specifically RetinaNet as the base object detection? Is the line 158-160, the sole reason?
Author response: Thanks for these insightful comments. Unlike other two-stage detectors like Faster R-CNN, the RetinaNet is a single-stage object detector that handles both object localization and classification at the same time. According to the literature, RetinaNet is currently one of the best one-stage object detection models that deliver superior performance in detecting both dense and small-scale regions of interest (ROIs). The architecture of the RetinaNet model encompasses a feature pyramid network that can detect ROIs at multiple scales. For these reasons, RetinaNet has been widely used in both natural image and medical image object detection applications.
Q6: In Figure 2, please mention the full form of conv. If it is convolution, then write as it is. Conv seems more casual.
Author response: Agreed and modified Figure 2 in the revised manuscript.
Q7: Line 163: 2.2.3. Ensemble of RetinaNet Models with Various Backbones; Line 183: 2.2.3. Loss functions and evaluation metrics. Both these sections have the same number. Maybe there is a mistake. Please check.
Author response: We regret the typos in the initial submission. The sub-sections are currently numbered in the revised manuscript.
Q8: In 2.2.3. Ensemble of RetinaNet Models with Various Backbones, it is mentioned that “Each model is trained for 80 epochs”, whereas in section 2.2.3. Loss functions and evaluation metrics it is mentioned “The training is performed for 128 epochs to minimize the categorical cross-entropy (CCE) loss.” Please address this more clearly.
Author response: Thanks. In Section 2.2.3, we discussed training the RetinaNet models with varying weight initializations. In Section 2.2.4.1 of the revised manuscript, we discussed training each of the modality-specific classifier backbones. The modality-specific classifier backbones are trained for 128 epochs. The RetinaNet models initialized with the modality-specific classifier backbones are trained for 80 epochs.
Q9: This section talks about the categorical cross-entropy (CCE) loss. But in equation (1), it is mentioned BCE. Please check if there is a mistake. Equation 1: Please mention in the text what is p, p’, Y, and Y’.
Author response: We regret the lack of clarity in the initial submission. We have modified the equation and the text (lines 209 – 218 ) in the revised manuscript.
Q10: I appreciate that the authors included the following section: 2.2.3.2. RetinaNet-Based Detection of Pneumonia-Consistent Findings. This is very specific to the study and expands on the importance of several parameters for image based classification. In equation (3) and (5) please mention what is c, and how it is related to the algorithm architecture.
Author response: We thank the reviewer for these appreciative words. The text and related equations are modified (lines 251 – 269) in the revised manuscript to convey clarity.
Q11: Please include a schematic architecture of the ensemble concept. It would help researchers form other fields to visualize the concept more easily. In Figure 4 and Figure 6, please mention what is AP? Is it mAP? Please be consistent with the image quality and size of Figure 4 and Figure 6.
Author response: Agreed and inserted a new Figure (Figure 3) in the revised manuscript. The quality of the figures is increased to 400 DPI, and the sizes are maintained consistently to suit journal requirements. In general terms, AP is calculated for each class and averaged to get the mAP. In this study, we have only one class, i.e., detecting bounding boxes consistent with pneumonia. So, both AP and mAP are the same. We have modified the figure legends to maintain consistency.
Q12: The authors mention several times the Modality specific based model, and even in the title. CXR image modality is first mentioned in section 3.1. Please mention it earlier, if possible, in the abstract.
Author response: Agreed. CXR modality-specific retraining is mentioned in lines 16 – 17 of the abstract and discussed in detail in lines 75 – 86 in the revised manuscript.
Q13: Are there any overfitting, or underfitting issues related to the results? How accurate will it work in the presence of other respiratory Illnesses caused from infections? Can it classify between the different infections?
Author response: Thanks for these insightful queries. We had sufficient data to train the CXR modality-specific classifier backbones and the RetinaNet models. Hence, we did not encounter overfitting issues. We used the DL models that were widely discussed to deliver superior performance with CXR visual recognition tasks in the literature. Hence, we did not encounter any underfitting issues. The flexibility and subsequent performance of the RetinaNet models are increased by using CXR modality-specific retrained classifier backbones. We used callbacks to store the model checkpoints and the training is stopped when the validation performance ceased to improve. The weights of the model that delivered a superior performance with the validation set are used to predict the test set. These steps ensured getting superior performance on the task under study. We believe that the CXR modality-specific backbone should help improve convergence and detection performance with other disease detection tasks since these models are trained on large-scale publicly available CXR collections that are acquired using various acquisition systems with diverse imaging protocols, from multiple institutions that help learn the variety towards normal lungs from those manifesting several other cardiopulmonary diseases. Unlike ImageNet weights, the models’ weights are converted specific to the CXR image modality, and the learned modality-specific knowledge can be transferred and fine-tuned for a related CXR visual recognition task. However, this assumption demands experimental evaluations and therefore remains a potential avenue for future research.
Reviewer 2 Report
This paper focuses on the issue of chest X-rays (CXRs) to screen for the infection. It has been well known that computer-aided detection methods using conventional deep learning (DL) models for identifying pneumonia-consistent manifestations in CXRs have demonstrated superiority over traditional machine learning approaches. This paper attempts to train a DL classifier on large collections of CXR images to develop a modality-specific model. Experimental results demonstrate that an ensemble of the top-3 performing RetinaNet models outperformed individual models in terms of the mean average. This research is interesting for the diagnosis research society. However, this paper has several limitations and the standard is not enough, and address the following items would result in a good paper,
1. The literature review is not thorough about the application and the contributions. To highlight the contributions, it suggests reorganizing the section of the related work with real applications. It is recommended to read more related works and consider discussing their application scenarios in the introduction and discussion, such as, A Cybertwin based Multimodal Network for ECG Patterns Monitoring using Deep Learning, A Multimodal Wearable System for Continuous and Real-time Breathing Pattern Monitoring During Daily Activity, etc.
2. The contribution of this paper is not clear. It suggests revising the contributions section and making these points clear and strong.
3. The quality of the Figures should be improved and readable for the readers.
4. Maybe it is better to discuss the possibility to improve the scope using deep learning to learn and optimize for online estimation in the introduction, for example, Improved recurrent neural network-based manipulator control with remote center of motion constraints: Experimental results, Towards Teaching by Demonstration for Robot-Assisted Minimally Invasive Surgery.
5. It is recommended to present in the first section so that it can highlight the specific scope of this article. The meaning of the assessment experiment should be highlighted.
6. Overall, proofreading is preferred. The current version is not written in a good and clear way. The English description should be improved and the grammar should be carefully addressed.
7. There should be a further discussion about the limitation of the current works, in particular, what could be the challenge for its related applications. To let readers better understand future work, please give specific research directions.
Author Response
This paper focuses on the issue of chest X-rays (CXRs) to screen for the infection. It has been well known that computer-aided detection methods using conventional deep learning (DL) models for identifying pneumonia-consistent manifestations in CXRs have demonstrated superiority over traditional machine learning approaches. This paper attempts to train a DL classifier on large collections of CXR images to develop a modality-specific model. Experimental results demonstrate that an ensemble of the top-3 performing RetinaNet models outperformed individual models in terms of the mean average. This research is interesting for the diagnosis research society. However, this paper has several limitations, and the standard is not enough, and address the following items would result in a good paper,
Author response: We render our sincere thanks to the reviewer for these valuable and insightful comments and for encouraging the resubmission of our manuscript. To the best of our knowledge and belief, we have addressed the queries of the reviewer in the revised manuscript.
Q1: The literature review is not thorough about the application and the contributions. To highlight the contributions, it suggests reorganizing the section of the related work with real applications. It is recommended to read more related works and consider discussing their application scenarios in the introduction and discussion, such as, A Cybertwin based Multimodal Network for ECG Patterns Monitoring using Deep Learning, A Multimodal Wearable System for Continuous and Real-time Breathing Pattern Monitoring During Daily Activity, etc.
Author response: Per the reviewer’s suggestion, the introduction section has been reorganized to include related works, the rationale for the study, and contributions of this study. We have also included the literature suggested by the reviewers as shown below:
- Qi, W.; Su, H. A Cybertwin Based Multimodal Network for ECG Patterns Monitoring Using Deep Learning. IEEE Trans. Ind. Informatics 2022, 3203, 1–9, doi:10.1109/TII.2022.3159583.
- Qi, W.; Aliverti, A. A Multimodal Wearable System for Continuous and Real-Time Breathing Pattern Monitoring during Daily Activity. IEEE J. Biomed. Heal. Informatics 2020, 24, 2199–2207, doi:10.1109/JBHI.2019.2963048.
Q2: The contribution of this paper is not clear. It suggests revising the contributions section and making these points clear and strong.
Author response: Thanks. We wish to reiterate our response to Q1.
Q3: The quality of the Figures should be improved and readable for the readers.
Author response: Per the reviewer’s suggestion, the quality of the figures is increased to 400 DPI, and the sizes are consistently maintained to suit journal requirements.
Q4: Maybe it is better to discuss the possibility to improve the scope using deep learning to learn and optimize for online estimation in the introduction, for example, Improved recurrent neural network-based manipulator control with remote center of motion constraints: Experimental results, Towards Teaching by Demonstration for Robot-Assisted Minimally Invasive Surgery.
Author response: Thanks. Per the reviewer’s suggestion, we have included the following literature in our study:
- Su, H.; Hu, Y.; Karimi, H.R.; Knoll, A.; Ferrigno, G.; De Momi, E. Improved Recurrent Neural Network-Based Manipulator Control with Remote Center of Motion Constraints: Experimental Results. Neural Networks 2020, 131, 291–299, doi:10.1016/j.neunet.2020.07.033.
- Su, H.; Mariani, A.; Ovur, S.E.; Menciassi, A.; Ferrigno, G.; De Momi, E. Toward Teaching by Demonstration for Robot-Assisted Minimally Invasive Surgery. IEEE Trans. Autom. Sci. Eng. 2021, 18, 484–494, doi:10.1109/TASE.2020.3045655.
Q5: It is recommended to present in the first section so that it can highlight the specific scope of this article. The meaning of the assessment experiment should be highlighted.
Author response: Agreed. Per the reviewer’s suggestion, the introduction section has been reorganized to include related works, the rationale for the study, and contributions of this study.
Q6: Overall, proofreading is preferred. The current version is not written in a good and clear way. The English description should be improved and the grammar should be carefully addressed.
Author response: Agreed. The revised manuscript has been proofread by a native English speaker and corrected for typos and grammatical errors.
Q7: There should be a further discussion about the limitation of the current works, in particular, what could be the challenge for its related applications. To let readers better understand future work, please give specific research directions.
Author response: Agreed. We have included the limitations of this study and scope for future research in the revised manuscript as shown below:
In this study, we demonstrated the combined benefits of training CXR image modality-specific models, using them as backbones in an object detection model, evaluating them in different loss settings, and constructing ensembles of the best-performing models to improve performance in a pneumonia detection task. We observed that both CXR image modality-specific classifier backbones and ensemble learning improved detection performance compared to the individual constituent models. This study, however, suffers from the limitation that we have only investigated the effect of using CXR modality-specific classifier backbones in a RetinaNet-based object detection model to improve detecting pneumonia-consistent findings. The efficacy of this approach in detecting other cardio-pulmonary disease manifestations is a potential avenue for future research. With the advent of high-performance computing and current advancements in DL-based object detection, future studies could explore the use of mask x-RCNN, transformer-based models, and other advanced detection methods [28–31] and their ensembles in improving detection performance. However, the objective of this study is not to propose a new objection detection model but to validate the use of CXR modality-specific classifier backbones in existing models to improve performance. As the organizers of the RSNA Kaggle pneumonia detection challenge have not made the blinded GT annotations of the test set publicly available, we are unable to compare our results with the challenge leaderboard. However, the performance of our method on a random split from the challenge-provided training set, where we sequester 10% of the images for testing, using 70% for training and 20% for validation, respectively, is markedly superior to the best performing method on the leaderboard. Additional diversity in the training process could be introduced by using CXR images and their disease-specific annotations collected from multiple institutions. Future research could explore novel model optimization methods and loss functions to further improve detection performance.
Round 2
Reviewer 2 Report
The authors have addressed all of my concerns. The current version can be accepted.